# European Pine Marten (*Martes martes*) as Natural Definitive Host of *Sarcocystis* Species in Latvia: Microscopic and Molecular Analysis

**DOI:** 10.3390/vetsci12040379

**Published:** 2025-04-17

**Authors:** Petras Prakas, Rasa Vaitkevičiūtė, Naglis Gudiškis, Emilija Grigaliūnaitė, Evelina Juozaitytė-Ngugu, Jolanta Stankevičiūtė, Dalius Butkauskas

**Affiliations:** 1Nature Research Centre, Akademijos 2, 08412 Vilnius, Lithuania; naglis.gudiskis@gamtc.lt (N.G.); emilija.grigaliunaite@gmail.com (E.G.); evelina.ngugu@gamtc.lt (E.J.-N.); dalius.butkauskas@gamtc.lt (D.B.); 2Agriculture Academy, Vytautas Magnus University, Studentų 11, Akademija, 53361 Kaunas, Lithuania; rasa.vaitkeviciute1@vdu.lt (R.V.); jolanta.stankeviciute1@vdu.lt (J.S.)

**Keywords:** *Sarcocystis*, Mustelidae, *Martes martes*, genetic identification, *cox1*, *ITS1*, definitive host, natural transmission

## Abstract

The pine marten (*Martes martes*) is a species of the family Mustelidae, widely distributed in the Baltic States. This predatory species can harbor a range of pathogens; however, its role in transmitting protozoan *Sarcocystis* parasites remains understudied. In the current investigation, we aimed to molecularly identify *Sarcocystis* species from intestinal samples of European pine martens from Latvia, focusing on the diagnosis of zoonotic and pathogenic ones. This is the first study to establish the richness of *Sarcocystis* species in the intestines of predators from Latvia. Overall, nine *Sarcocystis* species were identified: eight known ones, using cervids and livestock as their intermediate hosts, and a genetically new species, *Sarcocystis* sp. 25MmLV, closely related to *S. wenzeli*, which is known to be pathogenic for chickens. We report the first detection of *S. entzeorthi*, *S. hjorti*, *S. truncata*, and *Sarcocystis* sp. 25MmLV in European pine martens. Several of the *Sarcocystis* species that have been found can be pathogenic to farm animals or cervids. Furthermore, the detected *S*. *truncata* has been linked to food poisoning due to venison consumption.

## 1. Introduction

The Mustelidae family represents the most extensive and taxonomically diverse clade within the order Carnivora, encompassing five subfamilies and approximately sixty-seven extant species of terrestrial carnivorous or piscivorous mammals [1]. Mustelids exhibit remarkable ecological adaptability, thriving across diverse climates and seasonal environments. Their tendency to occupy large territories contributes to lower population densities than other carnivores [1,2]. Within the Mustelidae family, food habits vary widely among species, reflecting their adaptability to different environments. Many species hunt in burrows and crevices, while others have evolved specialized behaviors, such as climbing trees (e.g., martens) to pursue their diverse prey [1,3,4]. In Latvia, a total of nine species belonging to the Mustelidae family are found. These include weasels (*Mustela erminea*), stone martens (*Martes foina*), European minks (*Mustela lutreola*), European pine martens (*Martes martes*), least weasels (*Mustela nivalis*), badgers (*Meles meles*), polecats (*Mustela putorius*), American minks (*Mustela vison*), and otters (*Lutra lutra*) [5]. Historically, these animals have played an important role for humans, particularly in the fur trade and in controlling rodent populations [6,7]. The European pine marten is a key fur-bearing species found across the Baltic States. Although data from the Official Statistics of Latvia show a slight decline in the number of hunted animals from 2018 to 2025 (decreasing from 23 thousand to 22 thousand) [8], the precise estimated population of the European pine marten in Latvia remains uncertain. This uncertainty is compounded by the fact that, despite the overall decline in hunted animals, the population of European pine martens has been growing, likely due to reduced hunting pressure, which is largely attributed to a sharp decline in fur prices [9].

The European pine marten is known for its generalist and opportunistic feeding behavior, adapting its diet based on seasonal and local food availability. As a generalist omnivore, it likely occupies the most expansive dietary niche among species in the Mustelidae family [10]. Its diet shows considerable geographic and seasonal variation but commonly includes a variety of mammals, birds, fruits, invertebrates, amphibians, and reptiles [11,12,13]. The European pine marten interacts with a wide range of food sources, which exposes it to various parasites across different ecological niches. Given its diverse diet and ecological role, there is evidence suggesting that the European pine marten, although not extensively studied, may serve as a vector and potential reservoir for several pathogens [14]. To date, there have been documented cases of the European pine marten harboring a range of pathogens, including bacteria (e.g., *Leptospira* spp., *Yersinia* spp., *Salmonella* spp., *Helicobacter* spp.), protozoa (e.g., *Toxoplasma gondii*, *Neospora caninum*, *Hepatozoon* spp., *Sarcocystis* spp.), ectoparasites (e.g., *Demodex* spp., *Ixodes* spp.), and helminths such as trematodes (*Euryhelmis squamula*), cestodes (*Taenia martis*), and nematodes (*Crenosoma* spp., *Toxocara* spp., *Trichinella* spp., *Baylisascaris columnaris*) [9,15,16,17]. Bacterial pathogens like *Leptospira* spp., *Salmonella* spp., and *Yersinia* spp. are known to cause severe gastrointestinal, systemic, or vector-borne illnesses in humans and wildlife [18,19,20]. Also, the European pine marten contributes to the spread and circulation of important zoonotic protozoa and helminths, such as *Toxoplasma gondii*, *Eucoleus aerophilus*, *Capillaria hepatica*, *Angiostrongylus vasorum*, *Sarcocystis* spp., and *Trichinella* spp., in the environment [9,14]. Furthermore, zoonotic viruses such as SARS-CoV-2, Louping ill virus (LIV), and rotaviruses found in the European pine marten pose risks of cross-species transmission, potentially contributing to emerging infectious diseases [14].

The apicomplexan parasites of the genus *Sarcocystis* have zoonotic potential and can cause significant health issues in both livestock and wildlife, resulting in economic losses [21]. *Sarcocystis* spp. exhibit an obligate two-host life cycle, characterized by a prey–predator dynamic. The definitive host (DH), typically a carnivore or omnivore, becomes infected through the ingestion of mature sporocysts present in the muscle tissue of infected prey. Within the DH, *Sarcocystis* undergoes sexual reproduction, resulting in the release of sporocysts into the environment through fecal excretion, which subsequently contaminates food and water sources. These contaminated resources are consumed by animals serving as intermediate hosts (IHs). Upon ingestion, the sporocysts or oocysts are internalized by the IH, where *Sarcocystis* undergoes asexual reproduction, leading to the formation of sarcocysts mainly within the muscle tissue of the IH [22]. In recent years, the molecular characterization of *Sarcocystis* species has advanced significantly through the use of various genetic markers. Among these, the most commonly employed genetic regions in DH studies of *Sarcocystis* parasites include *18S* rRNA, *28S* rRNA, *ITS1*, and *cox1* [23,24,25,26]. Notably, the *cox1* gene has been shown to provide the highest resolution for differentiating *Sarcocystis* species infecting ungulates as IHs [27], whereas the *ITS1* region has proven to be the most informative marker for species identification when birds serve as IHs [28,29]. To date, over 200 distinct *Sarcocystis* species have been identified. Humans can become the DH of several *Sarcocystis* species by consuming beef or pork/wild boar meat [22,30,31]. Additionally, food poisoning has been linked to potential *Sarcocystis* spp. infections, particularly following the consumption of raw venison or horse meat, resulting in symptoms such as abdominal pain, vomiting, watery diarrhea, loss of appetite, and nausea [32,33]. *Sarcocystis* spp. primarily affect herbivorous IHs, causing significant tissue damage, increased mortality, and economic losses, while infections in carnivorous or omnivorous DHs are usually mild or asymptomatic, with occasional mild to chronic diarrhea. As such, *Sarcocystis* parasites pose a greater threat to IHs than to DHs [34,35].

*Sarcocystis* species exhibit varying degrees of pathogenicity across livestock, with significant implications for animal health and productivity. The pathogenicity of *Sarcocystis* depends on the species of parasite, the localization in the host, the dose of infection, and the immune status of the host [22]. *Sarcocystis* species transmitted by canids tend to be more pathogenic compared to those transmitted by other DHs [36]. *Sarcocystis cruzi* is the most pathogenic species in cattle, causing a range of clinical signs, including fever, anorexia, anemia, neuromuscular dysfunction, abortion, and mortality, with severity influenced by the infectious dose. Conversely, *Sarcocystis hirsuta* transmitted via felids is considered mildly pathogenic [37]. Typically, *Sarcocystis gigantea* results in few clinical signs in sheep, with economic losses mainly due to carcass condemnation [38]. However, occasionally, *Sarcocystis arieticanis* and *Sarcocystis tenella* can cause miscarriage or acute disease early in the infection, followed by chronic effects like reduced productivity [22]. In goats, *Sarcocystis capracanis* is the most pathogenic species, leading to fever, weight loss, miscarriage, and death, with survivors remaining unthrifty and prone to secondary infections [39]. *Sarcocystis bertrami*, which parasitizes horses, generally causes mild or no clinical signs [40]. In pigs, *Sarcocystis miescheriana* can induce severe symptoms, including weight loss, skin purpura, muscle tremors, miscarriage, and death, depending on the sporocyst load [41,42]. *Sarcocystis wenzeli* is potentially implicated as the etiological agent of meningoencephalitis in chickens [43,44]. Research on the pathogenicity of *Sarcocystis* spp. in wild ungulates of the Cervidae family are limited. Nevertheless, a study in Switzerland suggested *Sarcocystis hjorti* as the etiological agent of eosinophilic fasciitis in red deer [45], emphasizing the need for additional investigations to elucidate the pathogenic impact of various *Sarcocystis* species on wildlife health.

Recent studies employing molecular techniques have increasingly identified various mustelid species as natural DHs for several *Sarcocystis* spp. in Lithuania [46,47,48,49]. However, previous investigations tested a limited number of *Sarcocystis* species, emphasizing the diagnosis of parasite species that have certain IHs, such as cattle [49], cervids [47], birds [48], or livestock and cervids [46]. In the present study, we aimed to molecularly identify *Sarcocystis* spp. characterized by different IHs and DHs in the intestines of European pine marten collected in Latvia. It should be noted that we have focused on the detection of zoonotic *Sarcocystis* spp. and species pathogenic to farm animals and wildlife.

## 2. Materials and Methods

### 2.1. Sample Collection and Isolation of Sarcocystis spp. Sporocyst

Twenty European pine martens were legally hunted in the northern part of Latvia (main location: 57°20′33.1″ N, 25°25′21.4″ E) in February of 2023. Mustelids were collected in accordance with national and institutional guidelines from licensed third parties. The Cabinet of Ministers’ Hunting Law, Regulation No 421 (22 July 2014, Riga) classifies the European pine marten as an unlimited game animal and allows hunting from 1 October to 31 March. This study did not involve the purposeful hunting of animals. Intestinal samples from European pine martens were collected in collaboration with hunters and delivered to the Laboratory of Molecular Ecology at the Nature Research Centre (Vilnius, Lithuania). The material was preserved at –20 °C and subsequently used for the molecular examination of *Sarcocystis* species.

Fecal matter was expressed from each intestinal sample, after which the intestine was longitudinally incised. The mucosal surface was gently scraped with a scalpel, and the resulting material was suspended in 100 mL of distilled water. Sporocysts of *Sarcocystis* species were isolated from the full intestinal scrapings of each European pine marten following the protocol described by Verma et al. 2017 [50], with modifications as detailed in Šukytė et al. 2024 [51].

### 2.2. Molecular Analysis of Sarcocystis Species

DNA was extracted using the GeneJET Genomic DNA Purification Kit (Thermo Fisher Scientific Baltics, Vilnius, Lithuania), adhering to the provided protocol. The resulting purified DNA was stored at −20 °C for future molecular analyses.

*Sarcocystis* spp. were detected by amplifying partial *cox1* or *ITS1* sequences using nested PCR (nPCR). The *ITS1* region was used as a genetic marker for *Sarcocystis* species with avian IHs, while the *cox1* gene was employed for species infecting ungulates. The set of genus-specific (I step) and species-specific (II step) primer pairs used in this study is listed in Table 1. It is worth noting that the two most well-studied zoonotic *Sarcocystis* species (*S*. *hominis* and *S*. *suihominis*) were tested using species-specific primers in both rounds of nPCR. A total of 18 *Sarcocystis* species were examined. Of these, seven use members of the family Canidae as their DHs, including *S. arieticanis* (IH: wild and domestic sheep), *S. bertrami* (IH: horse), *S. capracanis* (IH: wild and domestic goats), *S. cruzi* (IH: cattle), *S. hjorti* (IH: Cervidae), *S. miescheriana* (IH: wild boars and pigs), and *Sarcocystis morae* (IH: Cervidae) [52,53,54,55,56,57,58]. Four species utilize placental predatory mammals as their DHs: *Sarcocystis anasi* (IH: ducks), *Sarcocystis albifronsi* (IH: geese), *Sarcocystis wenzeli* (IH: chicken), and *Sarcocystis rileyi* (IH: ducks) [59,60]. Meanwhile, Felidae serve as the DHs for *S. gigantea* (IH: wild and domestic goats) and *S. hirsuta* (IH: cattle) [57]. Two species employ Mustelidae as DHs, such as *Sarcocystis entzerothi* (IH: Cervidae) and *Sarcocystis truncata* (IH: Cervidae) [61,62]. The DHs of *Sarcocystis bovifelis* (IH: cattle) could be either Felidae or Mustelidae [63,64]. Additionally, zoonotic *Sarcocystis hominis* (IH: cattle) and *Sarcocystis suihominis* (IH: pigs and wild boars) use humans as their DHs [30,31]. The specificity of primers used was mainly confirmed in previous studies, except for hjor3/hjor4, which were tested in this study using *S*. *hjorti* DNA isolated from red deer (Table 1).

The first round of nPCR was conducted in a total reaction volume of 25 μL, comprising 12.5 μL of DreamTaq PCR Master Mix (Thermo Fisher Scientific, Vilnius, Lithuania), 0.5 μM of each primer (forward and reverse), 4 μL of extracted gDNA, and nuclease-free water to adjust the final volume. The PCR thermal cycling protocol commenced with an initial denaturation at 95 °C for 5 min, followed by 35 cycles consisting of denaturation at 94 °C for 35 s, annealing at a 55–69 °C (determined by the primer pair) for 45 s, and elongation at 72 °C for 55 s, concluding with a final extension step at 72 °C for 5 min. The second round of amplification was carried out in a 25 μL reaction mixture containing 12.5 μL of DreamTaq PCR Master Mix, 0.5 μM of each primer, 2 μL of the PCR product from the initial nPCR round, and nuclease-free water to adjust the final volume. The thermal cycling conditions were maintained identical to those in the initial round. To ensure the reliability of the amplification process, both positive and negative controls were included. Positive controls were prepared using genomic DNA extracted from *Sarcocystis* spp. sarcocysts identified in previous studies, while negative controls consisted of nuclease-free water instead of DNA.

The quality of the amplified nPCR products was checked using 1% agarose gel electrophoresis. Positive amplicons were purified with ExoI and FastAP (Thermo Fisher Scientific Baltics, Vilnius, Lithuania), following the manufacturer’s instructions. The purified DNA fragments were then sequenced using the same forward and reverse primers as in the nPCR. Sequencing reactions were carried out with the BigDye^®^ Terminator v3.1 Cycle Sequencing Kit and analyzed on a 3500 Genetic Analyzer (Applied Biosystems, Foster City, CA, USA), following the manufacturer’s guidelines. The resulting sequences were manually examined to ensure accuracy, ensuring there were no double peaks or poly signals.

### 2.3. Phylogentic Analysis

The *ITS1* and *cox1* sequences obtained in our work were deposited in GenBank with accession numbers PV364797, PV388221−PV388247, respectively. We used the nBLAST sequence similarity search algorithm [70] to compare sequences generated in the present work with those available in NCBI GenBank. BLAST analysis of the *ITS1* sequence obtained in this study revealed no significant similarity to any previously described *Sarcocystis* species, suggesting the presence of a genetically new species. Consequently, phylogenetic analysis was conducted to elucidate the phylogenetic relationships of this genetically new species. A multiple-sequence alignment was performed using a ClustalW algorithm incorporated into a MEGA11 v. 11.0.13 software [71]. The final alignment consisted of 27 sequences and 506 bp including gaps. The selection of nucleotide substitution model best fitting for a resulted alignment and a construction of a phylogenetic tree under a Bayesian inference was carried out using TOPALi v. 2.5 software [72]. The SYM nucleotide substitution model was chosen, and the tree was rooted on *Sarcocystis falcatula*. Bayesian analysis was performed in two runs, using half a million generations with a sample frequency of 10 and 25% burn-in value. We have used Microsoft Office suite for data visualization.

## 3. Results

### 3.1. Microscopical Examination of Sarcocystis spp. Sporocysts

Light microscopy (LM) analysis of intestinal scraping samples detected *Sarcocystis* spp. infection in 14 out of 20 (70.0%) European pine martens. Under LM, the detected free sporocysts were 8.0–12.9 × 6.1–8.6 µm (mean 10.6 × 7.1 µm; *n* = 70) in size (Figure 1). No sporulated oocysts or/and oocysts were detected in the intestinal samples analyzed.

### 3.2. Molecular Identification of Sarcocystis Species

Based on species-specific primers amplifying fragments of *cox1,* we identified eight known *Sarcocystis* species: *S*. *arieticanis*, *S*. *bertrami*, *S*. *capracanis*, *S*. *cruzi*, *S*. *entzerothi*, *S*. *hjorti*, *S*. *morae,* and *S*. *truncata* (Table 2). Other *Sarcocystis* species tested, including *S*. *anasi*, *S*. *albifronsi*, *S*. *bovifelis*, *S*. *giganta*, *S*. *hirsuta*, *S*. *hominis*, *S*. *rileyi*, *S*. *suihominis,* and *S*. *wenzeli,* were not detected. The comparison of sequences of the same species obtained in the current study showed minor differences (≤0.7%). By contrast, comparing our generated sequences with sequences of the same species available in GenBank, a significantly higher intraspecific variation was obtained, and sequence similarity values were in the range of 96.9–100% for *S*. *bertrami*, *S*. *capracanis*, *S*. *entzerothi*, *S*. *hjorti*, *S*. *morae,* and *S*. *truncata*. Meanwhile, our sequence of *S*. *cruzi* demonstrated 95.4–99.5% genetic similarity with those of *S*. *cruzi* and ≤90.4% genetic similarity with those of other *Sarcocystis* spp., and finally, our sequences of *S*. *arieticanis* shared 91.3–100% genetic similarity with *S*. *arieticanis* and ≤86.3% genetic similarity with those of other *Sarcocystis* spp. Thus, the calculated intraspecific and interspecific genetic variability values for all detected *Sarcocystis* species did not overlap, confirming the reliability of the species identification.

Furthermore, a genetically new species, named *Sarcocystis* sp. 25MmLV, was detected in the intestines of a single European pine marten by using internal SU1F/5.8SR2 and external AZVF1/AZVR1 primers (Table 1). The 461 bp *ITS1* sequence of *Sarcocystis* sp. 25MmLV displayed 92.3% genetic similarity to those of *Sarcocystis* sp. LT-2022 (OP970969–OP970970) isolated from the intestines of a European pine marten and American mink (*Neogale vison*) in Lithuania and to that of *Sarcocystis* sp. isolate Chicken-2016-DF-BR (MN846302) isolated from the brain tissues of chickens in Brazil; 91.2–92.1% genetic similarity to those of *S*. *wenzeli* (PQ192598, MT756994–MT756998) obtained from muscle tissues of chickens in China and from the feces of gray wolf (*Canis lupus*) from the USA; 91.4% genetic similarity to that of *Sarcocystis cristata* (MT676453) from muscle tissues of a great blue turaco (*Corythaeola cristata*) in the Central African Republic; and 90.9–91.3% genetic similarity to that of *Sarcocystis* sp. (OP490606–OP490609, OP490613–OP490614) isolated from the brains and muscles of chickens in Malaysia.

Based on the 461 bp *ITS1* sequence analyzed, *Sarcocystis* sp. 25MmLV was placed, with maximum support, in a cluster consisting of *S*. *wenzeli*, *S*. *cristata*, *S*. *anasi*, *S*. *albifronsi*, *S*. sp., and *S*. *chlorpopusae* (Figure 2), which form sarcocysts in the muscle or brain of various birds and are transmitted via placental predatory mammals of the order Carnivora [73]. *Sarcocystis* sp. 25MmLV was included in a different clade from that composed of *S*. *anasi*, *S*. *albifronsi,* and *Sarcocystis chloropusae*. A posterior probability support of 100% was given to group sequences of *S*. *wenzeli* (MT756996–MT756998) and *Sarcocystis* sp. isolated from chickens in China, Malaysia, and Brazil (MN846302, OP490606–OP490609, OP490613–OP490614) and sequences of *Sarcocystis* sp. isolated from the intestines of a European pine marten and an American mink from Lithuania (OP970969–OP970970). The fragment analyzed was not sufficiently phylogenetically informative to resolve the phylogenetic relationships between *Sarcocystis* sp. 25MmLV, *S*. *cristata,* and *Sarcocystis* species named *S*. *wenzeli* and isolated from the feces of gray wolf (PQ192598). However, based on BLAST analysis, the latter sequence (PQ192598) showed ≤91.9% genetic similarity compared to other available sequences of *Sarcocystis* spp. In summary, phylogenetic analysis confirmed that the *ITS1* sequence obtained in this work belongs to a genetically new *Sarcocystis* species.

### 3.3. Distribution of Sarcocystis Species Identified in Intestines of European Pine Martens

Of the 20 European pine martens examined, 13 were positive for *Sarcocystis* spp. by both microscopical and molecular methods. In two animals, no *Sarcocystis* spp. were identified; in four European pine martens, *Sarcocystis* spp. were only established by molecular examination; and finally, in one animal, *Sarcocystis* spp. sporocysts were observed, but no parasite species was diagnosed by PCR. The most commonly identified *Sarcocystis* species, *S*. *entzerothi,* was found in half of the samples, while the prevalence of other *Sarcocystis* spp. ranged from 5.0% to 25.0% (Figure 3a). The detection rate of *S*. *entzerothi* was significantly higher than compared to that of *S*. *hjorti*, *S*. *morae* (χ^2^ = 5.58, *p* < 0.05), *S*. *arieticanis*, *S*. *bertrami*, *S*. *truncata* (χ^2^ = 7.62, *p* < 0.01), *S*. *cruzi*, and *Sarcocystis* sp. 25MmLV (χ^2^ = 10.16, *p* < 0.01). In general, significantly more often *Sarcocystis* spp. with Cervidae as IHs (70.0%) were detected compared to those *Sarcocystis* spp. using Bovidae (25.0%, χ^2^ = 8.12, *p* < 0.01), Equidae (10.0%, χ^2^ = 15.00, *p* < 0.001), or Aves (5.0%, χ^2^ = 26.33, *p* < 0.001) as their IHs (Figure 3b). Based on molecular examination, 55.0% of the samples were positive for single *Sarcocystis* species, while co-infections with two or more *Sarcocystis* species were observed in 30.0% of the animals examined (Figure 3c). Two different *Sarcocystis* species were identified in 20.0% of European pine martens, and four and five different *Sarcocystis* species were detected in a single animal each.

## 4. Discussion

### 4.1. Identification of Sarcocystis spp. in Mustelidae as DH

*Sarcocystis* parasites predominantly are found as sarcocysts in the striated muscles of IHs, while in the DHs, they are detected in feces or the intestines as sporocysts or sporulated oocysts [22]. Traditionally, species differentiation has been based on variations in the sarcocyst wall structure within IHs [22,74]. However, this approach is not applicable in DH studies, as sporocysts and oocysts exhibit minimal morphological variation, lack distinct structural features, and barely differ in size, making species-level identification challenging [73,75]. To date, the majority of *Sarcocystis* species in DHs have been identified through transmission experiments [76,77,78]. However, the application of such experiments is significantly restricted worldwide due to ethical regulations [68]. Consequently, the analysis of fecal matter or intestinal scrapings is emerging as an alternative approach, enabling *Sarcocystis* identification without the need for live animal experimentation [79,80,81,82]. Using light microscopy, *Sarcocystis* sporocysts have been observed in 14 out of the 20 (70.0%) European pine martens in our study. Due to the limited data on *Sarcocystis* infection rates in mustelids, the only comparable data come from studies conducted in Lithuania, which have reported slightly lower infection rates: 47.6% [49], 53.0% [48], and 55.9% [46]. In other predatory mammals, infection rates range from 38.0% for red foxes [83] to 66.7% for raccoon dogs [68] and 39.4% for Pampas foxes [84].

Molecular techniques play a crucial role in identifying *Sarcocystis* species in DHs. A variety of genetic markers are used for precise *Sarcocystis* species differentiation. However, the genetic characterization of many *Sarcocystis* species remains limited, making it challenging to design species-specific primer pairs. For instance, this limitation is evident with *Sarcocystis linearis* and *Sarcocystis taeniata*, both of which employ cervids as IHs [85]. Thus, further molecular studies are necessary to expand the genetic database of *Sarcocystis* spp. Based on previous transmission experiments, it was confirmed that members of the Mustelidae family can serve as DHs for various *Sarcocystis* species with rodents as their IHs, including *Sarcocystis campestris* (IH: Sciuridae), *Sarcocystis citellivulpes* (IH: Sciuridae), *Sarcocystis muris* (IH: Muridae), *Sarcocystis putorii* (IH: Cricetidae), and *Sarcocystis undulati* (IH: Sciuridae) [22,86]. However, the actual diversity of *Sarcocystis* species within the Mustelidae family is likely much greater according to recent molecular reports [46,47,48,49], highlighting the need for further investigations. Furthermore, some of the sequence data available in GenBank may be unreliable due to potential misidentification or lack of morphological validation [87,88]. Therefore, caution is warranted when interpreting the results of genetic analyses based on such data.

### 4.2. Importance of European Pine Marten in Transmission of Zoonotic and Potentially Pathogenic Sarcocystis Species

The role of *Sarcocystis* spp. in zoonotic transmission is increasingly recognized; however, many aspects remain poorly understood. Historically, humans were considered the DH for only two *Sarcocystis* species, *S*. *hominis* and *S*. *suihominis* [22]. In 2015, a novel zoonotic species, *S. heydorni*, was described, though data on its geographical distribution remain scarce [89]. More recently, in 2024, *S. sigmoideus* was characterized, with its zoonotic potential demonstrated in 2025 [90,91]. Humans can acquire infection through the consumption of raw or undercooked beef containing sarcocysts of *S. hominis*, *S. heydorni*, and *S. sigmoideus* or pork/wild boar meat contaminated with *S. suihominis* [30,31,89,90]. In this study, no zoonotic *Sarcocystis* species were identified in the intestinal samples of European pine martens collected in Latvia. Testing for *S. hominis* and *S. suihominis* was performed following a prior report of *S. hominis* DNA detected in the intestines of a single Lithuanian pine marten [49]. These findings suggest that mustelids may act as accidental hosts of zoonotic *Sarcocystis*, likely acquiring the parasite through scavenging infected cattle meat or exposure to contaminated environments rather than serving as true reservoirs.

During this study, multiple *Sarcocystis* species were identified, including *S. bertrami, S. arieticanis, S. capracanis,* and *S. cruzi*, all of which are known to cause acute infections in farm animals. The aforementioned *Sarcocystis* species have long been thought to be distributed only by canine predators, but recent studies show that mustelids also contribute to the distribution of these species ([40,41,42,43], PS). However, data on the pathogenicity of *Sarcocystis* species infecting cervids remain scarce. While *S. hjorti* has been suggested as a causative agent of eosinophilic fasciitis in red deer [45], the broader impact of these parasites on their IHs is yet to be determined.

In parallel, increasing reports of *Sarcocystis*-related foodborne illnesses have raised concerns about their potential risk to human health. In Japan, *S. truncata* was implicated in a food poisoning outbreak linked to venison consumption [33,92,93]. The reported cases exhibited symptoms such as nausea, vomiting, and diarrhea within 24 h of ingestion, consistent with other known foodborne pathogens. However, the extent to which these species contribute to human poisoning remains unclear.

### 4.3. Composition of Sarcocystis Species in European Pine Marten

This is first study on the identification of *Sarcocystis* species in the intestines of predators in Latvia. A summarized overview of *Sarcocystis* species, along with their respective IHs and DHs identified by molecular methods in the intestines of the European pine marten and mustelids in general, is presented in Table 3. Notably, *S*. *entzeorthi*, *S*. *hjorti*, *S*. *truncata,* and *Sarcocystis* sp. 25MmLV have been identified in European pine marten for the first time worldwide. Additionally, this study provides the first molecular confirmation that a member of the Mustelidae family can serve as the natural DH of *S. hjorti*. However, in previous studies, *S. hjorti* was not screened as a potential species transmitted through mustelids using species-specific primers [46,47,48,49].

*Sarcocystis* species using cattle, sheep, goats, and horses as their IHs and canids as their DHs (*S*. *cruzi*, *S*. *arieticanis*, *S*. *capracanis*, *S*. *bertrami*) were previously confirmed in the intestines of the European pine marten in Lithuania [46,49] and in the present study, indicating that these *Sarcocystis* spp. can use both canids and mustelids for their natural transmission. Similarly, some of the examined *Sarcocystis* species (*S. hjorti, S. linearis,* and *S. morae*), originally characterized by a Cervidae–Canidae (IH-DH) life cycle, can also be transmitted by mustelids ([49], PS). By contrast, *S*. *miescheriana*, the type species of the genus *Sarcocystis* infecting pigs/wild boar [22], was not detected in the small intestines of the European pine marten. Phylogenetic analyses revealed that *Sarcocystis* species utilizing cervids and bovids as IHs and canids as DHs cluster into a single group. In contrast, species that infect equids and suids as IHs and canids as DHs form distinct phylogenetic lineages [94]. Thus, we propose the hypothesis that alternative transmission via mustelids is a relatively common occurrence within the phylogenetic cluster comprising *Sarcocystis* species that utilize cervids and bovids as IHs and canids as DHs.

Based on phylogenetic results, the eight *Sarcocystis* species described in Eurasia and using cervids as IHs—*Sarcocystis elongata*, *S*. *entzerothi*, *Sarcocystis japonica*, *S*. *matsuoae*, *Sarcocystis rangiferi*, *S*. *silva*, *Sarcocystis tarandi,* and *S*. *truncata*—are closely related and do not group with other *Sarcocystis* spp. using cervids and canids as their IHs and DHs, respectively [95,96]. Of these species, *S*. *elongata*, *S*. *entzerothi*, *S*. *japonica*, *S*. *silva,* and *S*. *truncata* were confirmed in the intestines of American minks from Lithuania [47]. In this study, two of the aforementioned *Sarcocystis* species (*S*. *entzerothi* and *S*. *truncata*) were also identified in European pine martens from Latvia, further supporting the hypothesis that *S. elongata*, *S. entzerothi*, *S. japonica*, *S. matsuoae*, *S. rangiferi*, *S. silva*, *S. tarandi,* and *S. truncata* are naturally transmitted by various mustelid species.

In the current study, we did not identify any *Sarcocystis* species characterized by a Bovidae–Felidae (IH-DH) life cycle. This is despite the occasional detection of *S. hirsuta* and the fact that the prevalence of *S. bovifelis* in mustelids from Lithuania, a neighboring country to Latvia, has been reported to be as high as 89.0% [46,49]. Further research is required to elucidate this phenomenon, with potential explanations including variations in the prevalence of *S. bovifelis* across different years or the relatively uncommon presence of this species in Latvia. In general, studies on *Sarcocystis* species in Latvia are limited and do not typically examine cattle as the IHs or predators as the DHs of these parasites [97]. In summary, European pine martens and other mustelid species play a significant role in the transmission of *Sarcocystis* species that use Bovidae, Cervidae, and Equidae as their IHs. However, many questions remain unanswered regarding which predators—canids or mustelids (or felids or mustelids in the case of *S. bovifelis*)—contribute more to the transmission of species such as *S. arieticanis, S. bertrami, S. capracanis, S. cruzi, S. hjorti,* and *S. morae*. It is important to note that previous research has primarily focused on the feces or intestines of one or a few predatory mammals within a single family [45,68,83]. Additionally, it should be considered that, in general, *Sarcocystis* species transmitted by canids cannot be transmitted by felids, and vice versa [22]. Therefore, future studies should focus on investigating predatory mammals from the Canidae and Mustelidae families, as well as from the Felidae and Mustelidae families, as potential DHs of the *Sarcocystis* species discussed herein, within a single, comprehensive study.

Using in silico primers designed to amplify *Sarcocystis* spp. infecting birds and transmittable via placental predatory mammals (*S*. *anasi*, *S*. *albifronsi*, *S*. *rileyi*, and *S*. *wenzeli*) [48], we identified a genetically novel *Sarcocystis* species closely related to *S*. *cristata* and *S*. *wenzeli* in single European pine marten specimen (Figure 2). Our phylogenetic analysis revealed that the *Sarcocystis* species named as *S. wenzeli*, isolated from the feces of a gray wolf (PQ192598) [81], does not cluster with other *S. wenzeli* isolates and may represent a distinct species. Additionally, *S. rileyi* and *Sarcocystis* sp. 25MmLV, which phylogenetic results suggest is most likely *S. wenzeli*, were detected in the intestines of 11.0% of 115 Lithuanian mustelids [48]. *Sarcocystis wenzeli* is pathogenic to chickens [43,44], while *S*. *rileyi* forms macroscopic sarcocysts resembling rice grains in the muscles of ducks, leading to economic losses due to contamination of hunted duck meat [48,59,98]. Thus, mustelids play a significant role in the transmission of important *Sarcocystis* species that infect birds consumed as human food.

## 5. Conclusions

In the present study, nine *Sarcocystis* species, *S*. *arieticanis*, *S*. *bertrami*, *S. capracanis*, *S*. *cruzi*, *S*. *entzerothi*, *S*. *hjorti*, *S*. *morae*, *S*. *truncata*, and *Sarcocystis* sp. 25MmLV, were identified in the intestinal samples of European pine martens from Latvia using molecular analysis. Some of the detected *Sarcocystis* species can be pathogenic to livestock or cervids. In addition, *S*. *truncata* has been associated with food poisoning due to the consumption of venison. Our study provides evidence that the European pine marten plays an important role as a DH for *Sarcocystis* species, using cervids and livestock as IHs. Moreover, further research is necessary to elucidate the ecological role of mustelids in the transmission dynamics of *Sarcocystis* spp.

## Figures and Tables

**Figure 1 vetsci-12-00379-f001:**
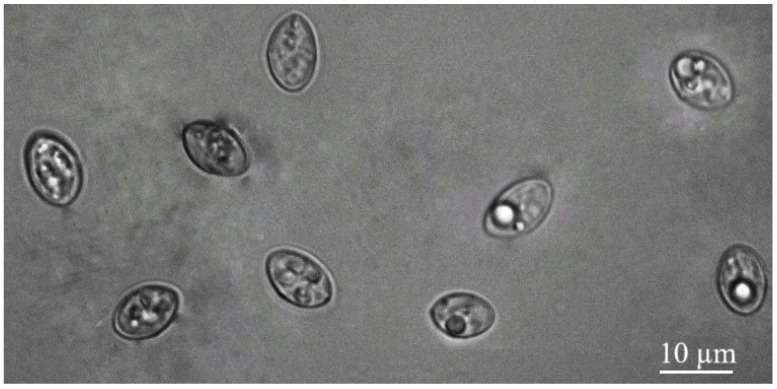
Sporocysts of *Sarcocystis* spp. from intestinal scrapings of European pine marten.

**Figure 2 vetsci-12-00379-f002:**
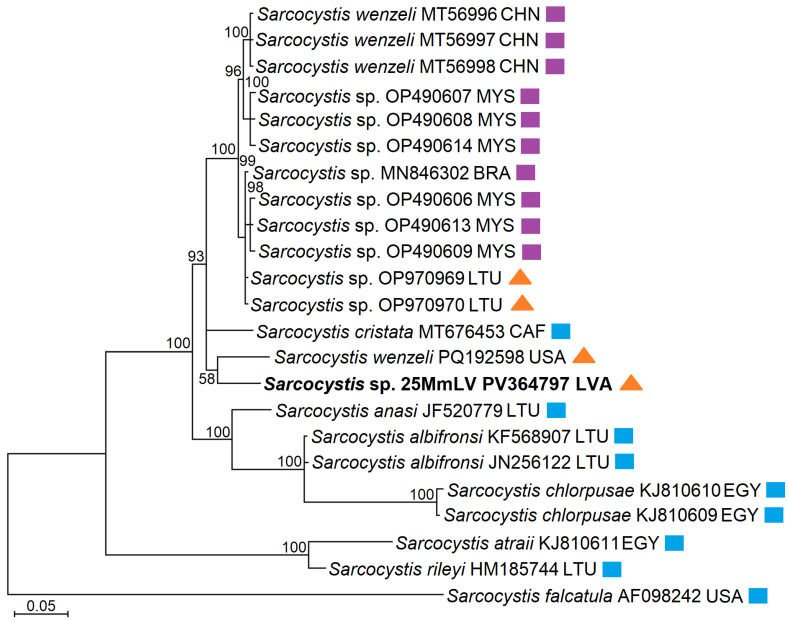
Bayesian phylogenetic tree based on *ITS1* sequences and showing the phylogenetic relationships of *Sarcocystis* sp. 25MmLV isolated from European pine marten from Lithuania. The phylogram was scaled according to branch length and rooted on *Sarcocystis falcatula*. GenBank accession numbers are given after *Sarcocystis* species names. The figures next to the branches display posterior probability values. Colored symbols indicate from which hosts the *Sarcocystis* species were isolated from; blue squares show that birds were the IHs, purple squares indicate that chickens were the IHs, and orange triangles show that Carnivora predators were the DHs. BRA—Brazil; CAF—the Central African Republic; CHN—China; EGY—Egypt; LTU—Lithuania; LVA—Latvia; MYS—Malaysia; USA—the United States of America.

**Figure 3 vetsci-12-00379-f003:**
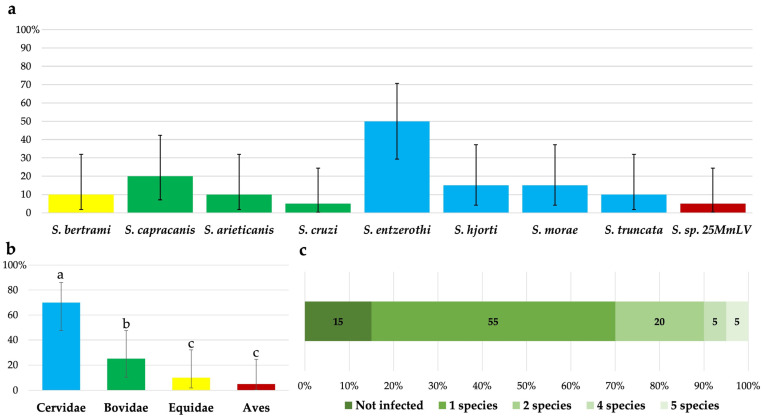
Trends in the distribution of *Sarcocystis* species in analyzed intestinal samples of European pine marten from Latvia. (**a**) The prevalence of *Sarcocystis* species in the samples examined. (**b**) The prevalence of *Sarcocystis* species grouped by their IH. a > b (*p* < 0.01) and a > c (*p* < 0.001). (**c**) Percentage distribution of the number of *Sarcocystis* species identified per sample.

**Table 1 vetsci-12-00379-t001:** Nested PCR primer pairs used in this study.

Sarcocystis Species	Primer Name	Step	Sequence	IH/DH
*Sarcocystis* spp.	SF1 [27]	I	ATGGCGTACAACAATCATAAAGAA	Ungulate/Canid
	SsunR3 [65]		CCGTTGGWATGGCRATCAT	
*S*. *arieticanis*	Arieticanis7F [57]	II	TAATTTCCTCGGTACTGTACTGTTTG	Sheep/Canid
	Arieticanis7R [57]		TACTTACGCATTGCGATATTACG	
*S*. *capracanis*	V2ca3 [57]	II	ATACCGATCTTTACGGGAGCAGTA	Goat/Canid
	V2ca4 [57]		GGTCACCGCAGAGAAGTACGAT	
*S*. *cruzi*	V2cr3c [66]	II	TCCAAGTACACGGCATTATTTACC	Cattle/Canid
	V2cr4 [66]		AAACTACTTTACTGCCTACGGTACTC	
*S*. *bertrami*	V2ber7 [66]	II	CCCCACTCAGTACGAACTCC	Horse/Canid
	V2ber8 [66]		ACTGCGATATAACTCCAAAACCA	
*S*. *miescheriana*	V2mie5 [66]	II	TCCTCGGTATTAGCAGCGTACTG	Pig/Canid
	V2mie6 [66]		ATTGAAGGGCCACCAAACAC	
*S*. *hjorti*	V2hjo3 [PS]	II	GGGCCATCATATTTACAGCATT	Cervid/Canid
	V2hjo4 [PS]		GAAAACTACCCTGCCGCCTA	
*S*. *morae*	V2mor1 [57]	II	GTGTGCTTGGATCGGTCAAC	Cervid/Canid
	V2mor2 [57]		GCCGAATACCGGCTTACTTC	
*Sarcocystis* spp.	SF1 [27]	I	ATGGCGTACAACAATCATAAAGAA	Ungulate/ Felid-Mustelid
	SkatR [49]		CAGGCTGAACAGHABTACGA	
*S*. *bovifelis*	V2bo3 [49]	II	ATATTTACCGGTGCCGTACTTATGTT	Cattle/Felid-Mustelid
	V2bo4 [49]		GCCACATCATTGGTGCTTAGTCT	
*Sarcocystis* spp.	SF1 [27]	I	ATGGCGTACAACAATCATAAAGAA	Ungulate/Felid
	SkatR2 [46]		GCTGAACAGTATTACGAATGATATG	
	SkatV1 [46]	II	AGTTTGGCGCTGCCGTAG	
	SkatV2 [46]		TCAGGGTGCCCGAAGAAC	
*S*. *gigantea*	V2gig3 [49]	II	CAGCAAGTACCAAGTTCTGTACGTC	Sheep/Felid
	V2gig4 [65]		GGTGCCGAGTACCGAGATACAT	
*S*. *hirsuta*	V2hi7 [67]	II	GCACCGTAATATTTCAGGGATGT	Cattle/Felid
	V2hi8 [67]		AACCTGCTTGCCGGAGTAAGTA	
*Sarcocystis* spp.	SF1 [27]	I	ATGGCGTACAACAATCATAAAGAA	Ungulate/ Felid-Mustelid
	SelniaiR [47]		AAATAYCTTRGTGCCCGTAG	
*S*. *entzerothi*	GsSentF2 [41]	II	AACTTCCTGGGTACCGCCATT	Cervid/Mustelid
	GsSentR2 [41]		TATGGCAATCATAATGGTTACAGCA	
*S*. *truncata*	GsStruF3 [41]	II	TTTGTTGGTTCCGTAAGTGCCTAT	Cervid/Mustelid
	GsStruR3 [41]		GGTATCAACCTCCAATCCAACTGT	
*S*. *hominis*	GaHoEF [64]	I	TCTCTGGTTTTGGTAACTACTTCGT	Cattle/Human
	GaHoER [64]		CAGACACTGGGATATAATACCGAAC	
	GaHoEF2 [64]	II	CATTGGCTGGACTCTCTATGCT	
	GaHoER2 [64]		AAATATCGGCAGGGTAATTATCAA	
*S*. *suihominis*	V2su5 [68]	I	CAACGTGTACTTTACCATGCAC	Pig/Human
	V2su6 [68]		AGCCGGGCAGAATCAGAATA	
	V2su7 [68]	II	GTATGGCTAATCCACTCCGTAA	
	V2su8 [68]		GCATCATAAAAACCAAAGTTGAG	
*Sarcocystis* spp.	SU1F [69]	I	GATTGAGTGTTCCGGTGAATTATT	Birds-Mammals/Birds-Mammals
	5.8SR2 [69]		AAGGTGCCATTTGCGTTCAGAA	
*S*. *anasi*/*S*. *albifronsi*/*S*. *wenzeli*	AZVF1 [48]	II	TCAAAACGTCCAAATAATGGTAT	
	AZVR1 [48]		ACACATTCCTACTGCCTTCCAC	
*S*. *rileyi*	GsSrilF2 [48]	II	ACGTTGTTCTATATTATGTGACCATT	
	GsSrilR2 [48]		TACTATAGAGGTGAAAGGGAGGTGA	

PS—present study.

**Table 2 vetsci-12-00379-t002:** Comparison of the *cox1* sequences obtained in this work with each other and with the sequences available in GenBank.

Sarcocystis Species	GenBank Accession Numbers	Intraspecific Genetic Similarity ^1^	Interspecific Genetic Similarity ^2^
*S*. *arieticanis*	PV388221-PV388222	100% (91.3–100% ^3^)	85.5–86.3% *Sarcocystis hircicanis*
*S*. *bertrami*	PV388223-PV388224	99.7% (97.3–100%)	81.3–82.2% *Sarcocystis asinus*
*S*. *capracanis*	PV388225-PV388228	99.3–100% (97.5–99.3%)	91.0–93.6% *S*. *tenella*
*S*. *cruzi*	PV388229	N/A (95.4–99.5%)	89.9–90.4% *Sarcocystis levinei*
*S*. *entzerothi*	PV388230-PV388239	99.7% (99.3–100%)	91.3–91.6% *Sarcocystis matsuoae*
*S*. *hjorti*	PV388240-PV388242	100% (98.0–100%)	92.6–94.1% *Sarcocystis pilosa*
*S*. *morae*	PV388243-PV388245	100% (96.9–100%)	83.6–84.9% *Sarcocystis cervicanis*
*S*. *truncata*	PV388246-PV388247	100% (99.1–100%)	95.3–96.5% *Sarcocystis silva*

^1^ The first figures show values obtained comparing sequences of the same species determined in this study, while figures in parentheses show a comparison of our sequences this those of the same species available in GenBank. ^2^ Comparing our sequences with those of most closely related species by the BLAST tool. ^3^ A similarity of 97.5–100% was obtained by omitting MH413047-MH413048 sequences from Egypt. N/A not applicable.

**Table 3 vetsci-12-00379-t003:** Molecular identification of *Sarcocystis* species characterized by different IHs and DHs in the intestines of mustelids and the European pine marten.

Sarcocystis Species *	Intermediate Host	Typical Definitive Host	Previously Found in Mustelidae (in European Pine Marten), +/−	Identified in Current Work, +/−
*S*. *arieticanis*	Bovidae	Canidae	+ (+) [46]	+
*S*. *capracanis*	Bovidae	Canidae	+ (+) [46]	+
*S. cruzi*	Bovidae	Canidae	+ (+) [49]	+
*S*. *bertrami*	Equidae	Canidae	+ (+) [46]	+
*S*. *miescheriana*	Suidae	Canidae	− (−) [46]	−
*S*. *hjorti*	Cervidae	Canidae	− (−)	+
*S*. *morae*	Cervidae	Canidae	+ (+) [46]	+
*S*. *bovifelis*	Bovidae	Felidae	+ (+) [49]	−
*S*. *gigantea*	Bovidae	Felidae	− (−) [46]	−
*S*. *hirsuta*	Bovidae	Felidae	+ (+) [46]	−
*S*. *entzerothi*	Cervidae	Mustelidae	+ (−) [47]	+
*S*. *truncata*	Cervidae	Mustelidae	+ (−) [47]	+
*S*. *hominis*	Cattle	Humans	+ (+) [49]	−
*S*. *suihominis*	Pig/wild boar	Humans	− (−)	−
*Sarcocystis* spp.	Aves	Canidae, Mephistidae	+ ^1^ (+ ^2^) [48]	+ ^3^

* If any of the species listed in the row have been detected, a + sign is displayed. ^1^
*S. rileyi, Sarcocystis* sp. LT-2022, ^2^
*Sarcocystis* sp. LT-2022, ^3^
*Sarcocystis* sp. 25MmLV.

## Data Availability

The sequences obtained in our work are available in the GenBank database under accession numbers PV364797 (*ITS1*) and PV388221–PV388247 (*cox1*).

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
