# Peer review of "European Pine Marten (Martes martes) as Natural Definitive Host of Sarcocystis Species in Latvia: Microscopic and Molecular Analysis"

_vetsci, 2025, doi:10.3390/vetsci12040379_

Round 1

Reviewer 1 Report

Comments and Suggestions for Authors

157: should describe how much feces or how the sporocysts were concentrated prior to extraction. This is important for trying to replicate this in other studies of sarcocystis in mustelids and other carnivores

213  it says the ITS sequence cannot be assigned to any sarcocystis species

197: did they really have positive controls for all species listed?

228: were there other apicomplexans detected?  Were all the sporocysts the same size? That is surprising given all of the species detected?

247: Some of the species are not common; can the authors comment of how reliable/bona fide are the available genbank isolates?

252 and similar to above: why omit the sequences from Egypt?

Paragraph with 335:  I’m a little confused with there are both so many genes available for species identity but then cox1 is used here. In general cox1 is more hypervariable compared to 18S for example.  In particular, if one uses a single gene to conduct phylogeny, there are more likely to be errors whereas phylogenetic trees can be constructed using concatenated results from multiple genes which makes the analysis more robust. Ideally they would take this approach but they should at least discuss this point

I guess I would welcome the authors to be a little introspective of the limits of the genbank specimens. They say the identification of species based on host transmission is limited but seem quite confident that the sequencing result is definitive.  They should at least comment on the possible error introduced by the genbank sequence if it is not definitely proven since these are not all well known species

358: regarding humans as intermediate hosts of Sarcocystis- wouldn’t they have to consume sporocysts in feces? If they are ingesting the beef/pork/etc, they would be the definitive host.  Should clarify the role of the human in these life cycles (DH vs IH)

Are there no species that martens acquire via eating rodents? While I’m not totally surprised about cattle/cervid species, I would expect their diet to have more small mammals. Perhaps part of the discussion should address the known diets of martens there.  While this is probably outside the scope of the current study, there are methods for using PCR/sequencing to determine dietary content and it would be interesting to correlate those findings with the sarcocystis findings.

I’m impressed by the wide variety of species discovered by sampling a relatively small number of animals. I suggest the authors discuss if this is expected vs unexpected. In our hands finding sporocysts is fairly rare, although we examine very few mustelids. Was there any kind of power analysis done based on prevalence estimates to arrive at sampling ~20 individual animals?  If they sampled more are they likely to find more sarcocystis species or is this about what they expect to find?

Author Response

Point 1: 157: should describe how much feces or how the sporocysts were concentrated prior to extraction. This is important for trying to replicate this in other studies of Sarcocystis in mustelids and other carnivores.

Response: We have included the necessary reference detailing the protocol for the isolation and concentration of sporocysts to enable reproducibility by the scientific community. To avoid redundancy and repetition, we chose not to rewrite the entire methodology in this study, as the original protocol is thoroughly described and can be accessed through the cited references.

Point 2: 213 it says the ITS sequence cannot be assigned to any Sarcocystis species.

Response: Thank you for your valuable comment. What we intended to convey is that the ITS1 sequence obtained in this study represents a genetically distinct Sarcocystis species, which could not be assigned to any previously described species based on BLAST analysis. To clarify this point, we have revised the corresponding section in the Methods (Lines 220–224) to make our approach and interpretation more explicit.

Point 3: 197: did they really have positive controls for all species listed?

Response: We did have the majority of the necessary positive controls for this study, which were obtained during our earlier research in the laboratory. The missing positive controls of S. hominis, S. suihominis and S. gigantea were acquired through collaboration with other scientists in the field of Sarcocystis.

Point 4: 228: were there other apicomplexans detected?  Were all the sporocysts the same size? That is surprising given all of the species detected?

Response: Thank you for your remark. No other Apicomplexan parasites were detected through light microscopy analysis. As indicated in Line 237, the observed free sporocysts did vary in size. Thus, while variation in sporocyst size was observed, it is not sufficient to accurately identify the corresponding Sarcocystis species.

Point 5: 247: Some of the species are not common; can the authors comment of how reliable/bona fide are the available Genbank isolates?

Response: Thank you for your observation. We included only reliable and well-validated sequences, omitting those that lacked verification or morphological confirmation. Specifically, some sequences attributed to Sarcocystis bertrami and Sarcocystis tenella were excluded due to the absence of supporting data or inconsistent identification in GenBank.

Point 6: 252 and similar to above: why omit the sequences from Egypt?

Response: When comparing Sarcocystis arieticanis sequences obtained in this study with those available in the GenBank database, we observed an intraspecific sequence similarity ranging from 91.3% to 100%. Notably, the lowest similarity values were primarily influenced by sequences originating from Egypt, which exhibited unusually high divergence. When Egyptian sequences were excluded, the intraspecific similarity among the remaining S. arieticanis sequences increased to 97.5-100%. This suggests that the Egyptian sequences may represent either regional variants with elevated genetic divergence or potentially misidentified entries in the GenBank database. However, please not that in the table similarity values are presented including Egyptian sequences.

Point 7: Paragraph with 335:  I’m a little confused with there are both so many genes available for species identity but then cox1 is used here. In general cox1 is more hypervariable compared to 18S for example.  In particular, if one uses a single gene to conduct phylogeny, there are more likely to be errors whereas phylogenetic trees can be constructed using concatenated results from multiple genes which makes the analysis more robust. Ideally, they would take this approach but they should at least discuss this point.

Response: We are grateful for your insightful feedback. However, the cox1 gene is currently regarded as the most suitable marker for species-level identification of Sarcocystis spp. infecting ungulates as intermediate hosts (Gjerde, 2013; https://doi.org/10.1016/j.ijpara.2013.02.004). In contrast, for Sarcocystis species that utilize birds as intermediate hosts, the ITS1 region has proven to be the most informative genetic marker, offering greater discriminatory power than more conserved regions such as 18S rRNA or 28S rRNA (Prakas et al., 2025; https://doi.org/10.1016/j.vetpar.2025.110413). Given that the genetically distinct Sarcocystis species identified in our study was detected using the ITS1 region, the phylogenetic tree was constructed exclusively based on this marker to accurately resolve its evolutionary relationships in the context of other avian Sarcocystis species.

Point 8: I guess I would welcome the authors to be a little introspective of the limits of the genbank specimens. They say the identification of species based on host transmission is limited but seem quite confident that the sequencing result is definitive.  They should at least comment on the possible error introduced by the Genbank sequence if it is not definitely proven since these are not all well-known species.

Response: We acknowledge that some of the sequence data available in GenBank may lack proper verification, particularly in cases where species identification is based solely on molecular data without morphological confirmation. This is especially important when determining new definitive hosts of Sarcocystis species, as reliance on unverified sequences may lead to misinterpretation. Therefore, we have added a brief discussion addressing this concern in Lines 359–362 to highlight the need for caution when interpreting such genetic data.

Point 9: 358: regarding humans as intermediate hosts of Sarcocystis-wouldn’t they have to consume sporocysts in feces? If they are ingesting the beef/pork/etc, they would be the definitive host. Should clarify the role of the human in these life cycles (DH vs IH)

Response: To clarify the potential role of humans as IHs for Sarcocystis species, infection may occur through the ingestion of water contaminated with sporocysts or oocysts. Potential sources of Sarcocystis infection in IHs are also outlined in Lines 98-99. However, as this study does not address cases in which humans have been incidentally identified as IHs for Sarcocystis species (e.g., S. nesbitti), we prefer not to introduce additional data on this topic. Including such information could lead to confusion and undermine the focus on the significance of Sarcocystis species in humans as DHs.

Point 10: Are there no species that martens acquire via eating rodents? While I’m not totally surprised about cattle/cervid species, I would expect their diet to have more small mammals. Perhaps part of the discussion should address the known diets of martens there.  While this is probably outside the scope of the current study, there are methods for using PCR/sequencing to determine dietary content and it would be interesting to correlate those findings with the Sarcocystis findings.

Response: Thank you for your suggestion. As noted in Lines 344–348, there are Sarcocystis species for which members of the family Mustelidae may serve as DHs. The diet of mustelids, particularly the European pine marten, includes a notable proportion of rodents. Despite this, the genetic characterization of Sarcocystis species in rodents remains limited, which hampers the development of species-specific primer pairs and the identification of such species in DHs. While transmission experiments could help identify which rodent species could parasitize in mustelids, such studies are constrained by ethical considerations. In this study, we focused exclusively on Sarcocystis species with IHs among farm and wild animals. Nevertheless, future investigations into Sarcocystis species involving mustelids as potential DHs would be both valuable and informative, potentially broadening our understanding of the diversity of Sarcocystis in these hosts.

Point 11: I’m impressed by the wide variety of species discovered by sampling a relatively small number of animals. I suggest the authors discuss if this is expected vs unexpected. In our hands finding sporocysts is fairly rare, although we examine very few mustelids. Was there any kind of power analysis done based on prevalence estimates to arrive at sampling ~20 individual animals?  If they sampled more, are they likely to find more Sarcocystis species or is this about what they expect to find?

Response: Thank you for your insightful comment. Based on our experience, the prevalence of infection in mustelids typically exceeds 50%. However, the number of sporocysts observed per unit area is significantly lower compared to canids. Similar trends have been reported in other studies. It is evident that examining a larger number of animals would likely result in a greater diversity of detected species, especially on rare ones. Nonetheless, this approach is limited by the practical challenges associated with collecting sufficient numbers of suitable host animals, which is not always feasible. It should be noted that hinting of mustelids is now significantly decreasing in the  Baltic States as the need to use the fur of these animals is decreasing.

Reviewer 2 Report

Comments and Suggestions for Authors
  1. The introduction extensively describes the species and living habits of the Mustelidae family, as well as the dietary characteristics of the European pine marten and the pathogens it harbors. However, it lacks relevant research backgroudregarding the molecular identification and phylogenetic analysis of Apicomplexan Sarcocystis It is recommended to make appropriate adjustments.
  2. Why was the phylogenetic tree constructed based on the ITS1 gene? The article proposed that the cox1 gene is a reliable typing locus. What role did cox1 play in the molecular identification in this study?
  3. In the methods section, the data analysis method for "The Distribution of Sarcocystis SpeciesIdentified in Intestines of European Pine Martens" needs to be supplemented.
  4. What is the data source of the positive rate of Sarcocystisparasites in the intermediate hosts presented in the results?

Author Response

Point 1: The introduction extensively describes the species and living habits of the Mustelidae family, as well as the dietary characteristics of the European pine marten and the pathogens it harbors. However, it lacks relevant research background regarding the molecular identification and phylogenetic analysis of Apicomplexan Sarcocystis. It is recommended to make appropriate adjustments.

Response: We have added information regarding the molecular identification of Sarcocystis species and the genetic markers commonly used for this purpose in both the Introduction (Lines 101-108) and Methods (Lines 171-172) sections.

Point 2: Why was the phylogenetic tree constructed based on the ITS1 gene? The article proposed that the cox1 gene is a reliable typing locus. What role did cox1 play in the molecular identification in this study?

Response: Since a genetically new Sarcocystis species was identified based on the ITS1 region, the phylogenetic analysis was accordingly performed using the same marker to ensure consistency and accurate resolution of species-level relationships. It is explained in methodological part of the manuscript.

Point 3: In the methods section, the data analysis method for "The Distribution of Sarcocystis Species Identified in Intestines of European Pine Martens" needs to be supplemented.

Response: We have added the necessary information in Lines 233-234.

Point 4: What is the data source of the positive rate of Sarcocystis parasites in the intermediate hosts presented in the results?

Response: Thank you for your comment. If we understand your question correctly, the positive rate of Sarcocystis parasites in IHs , as presented in Figure 3, was calculated as the mean infection rate across the various Sarcocystis species that utilize cervids as their IHs. Specifically, we averaged the infection rates of all Sarcocystis species known to infect cervids and divided this by the total number of Sarcocystis species associated with cervids as IHs. Information regarding the IH groups (e.g., Cervidae) and the host species within these groups can be found in Sections 2.2 (Methods) and 4.3 (Discussion). The data for these sections were compiled from existing scientific literature.